# Impact of Aging and Visual Input on Postural Stability in Dogs: Insights from Center-of-Pressure Analysis

**DOI:** 10.3390/s25051300

**Published:** 2025-02-20

**Authors:** Christiane Lutonsky, Christian Peham, Nadja Affenzeller, Masoud Aghapour, Julia Wegscheider, Alexander Tichy, Barbara Bockstahler

**Affiliations:** 1Section of Physical Therapy and Rehabilitation, Small Animals Surgery, Clinical Centre for Small Animal Health and Research, Clinical Department for Small Animals and Horses, University of Veterinary Medicine, 1210 Vienna, Austria; nadja.affenzeller@vetmeduni.ac.at (N.A.); masoud.aghapour@vetmeduni.ac.at (M.A.); barbara.bockstahler@vetmeduni.ac.at (B.B.); 2Movement Science Group, Clinical Centre for Equine Health and Research, Clinical Department for Small Animals and Horses, University of Veterinary Medicine, 1210 Vienna, Austria; christian.peham@vetmeduni.ac.at; 3Behavioral Medicine, Clinical Centre for Small Animal Health and Research, Clinical Department for Small Animals and Horses, University of Veterinary Medicine, 1210 Vienna, Austria; 4Department of Biological Sciences and Pathobiology, Platform for Bioinformatics and Biostatistics, Centre of Biological Sciences, University of Veterinary Medicine, 1210 Vienna, Austria; alexander.tichy@vetmeduni.ac.at

**Keywords:** posturography, aging, center of pressure, blindfolding, dogs

## Abstract

This study investigates the impact of visual input and aging on postural stability (PS) in dogs by analyzing center-of-pressure (COP) parameters during static posturography under sighted (EO) and blindfolded (EC) conditions. Twenty adult (<50% of fractional lifespan) and 20 senior (>75% of fractional lifespan) dogs, free from orthopedic, neurological, or visual impairments, were assessed using a pressure measurement plate. While no significant differences were found between adult and senior dogs under standard EO conditions, blindfolding revealed age-related disparities. Senior dogs exhibited significantly higher craniocaudal displacement and support surface values compared to adult dogs, indicating a greater reliance on visual input for sagittal stability. Conversely, adult dogs exhibited a reduction in postural sway during EC conditions, indicating an adaptive shift toward greater reliance on somatosensory input. These findings highlight diminished sensory integration and adaptability in senior dogs, correlating with aging-related declines in proprioception and sensory processing. This research underscores the critical role of vision in canine PS, particularly in older individuals, and emphasizes the need for targeted interventions, such as balance training, to enhance sensory integration and mitigate fall risk in aging dogs. Future studies should explore dynamic and multimodal challenges to further elucidate compensatory mechanisms.

## 1. Introduction

Postural stability (PS) is the ability to maintain the body’s center of mass within the base of support (BOS), whether stationary or during motion [1,2,3]. It involves the complex integration of sensory inputs (visual, vestibular, and proprioceptive) and motor responses to adjust body position and orientation [1,4]. Effective PS reflects the body’s capacity to resist or recover from disturbances, ensuring physical stability and efficient movement [5].

The BOS, defined as the area enclosed by contact points between the body and the standing surface, plays a fundamental role in PS. In humans, increasing the BOS through a wider stance enhances passive stability by increasing the distance between the center of mass and the mediolateral limits of the BOS [6]. Conversely, narrowing the BOS, as observed during single-leg [7,8,9] or tandem stance (heel-to-toe position) [10,11,12], increases the challenge of maintaining PS.

The functional BOS, defined as the area used to maintain balance, may differ from the total BOS [13]. If the center of mass exceeds the functional BOS, protective steps are required to avoid a fall [14]. The functional BOS tends to decrease with age, contributing to increased fall risk in human patients, but can be improved with balance training [15,16].

In humans, falls are a major health concern, often leading to severe injuries due to declining physical function. As a result, fall prevention is a key area of focus in both clinical research and intervention strategies [17,18]. Similarly, aging dogs face an increased risk of falls due to muscle weakness, balance deficits, and age-related conditions such as musculoskeletal and neurological disorders. However, because dogs are quadrupeds and have a lower center of gravity, they are less likely to sustain serious injuries from falls compared to humans. Despite this reduced injury risk, maintaining PS in senior dogs is still crucial, as falls can negatively impact their overall health and mobility. Therefore, assessing and addressing balance impairments in aging dogs are important aspects of veterinary care to help preserve their quality of life [19].

The COP, a key parameter for studying PS in humans [9,20,21,22,23,24,25,26] and animals [27,28,29,30,31,32,33,34,35,36,37,38,39,40], represents the vertical projection of the center of mass and reflects the dynamic interaction of ground reaction forces during stance and motion [25]. The following COP parameters are used in canine research: the COP displacement in the craniocaudal (CCD) and mediolateral (MLD) directions, the statokinesiogram length (L) defined as the length of the line connecting the coordinates of the COP trajectory at each timeframe, the support surface (SS) which is the area of the ellipse containing 90% of all COP points, and the average COP speed (AS) [30,39]. As stated above, the diameters of the BOS influence postural strategies [9,10,11,12,13,14,15], which is reflected in COP parameters. Thus, ensuring a comparable BOS during test conditions and across groups is essential when analyzing COP parameters in static measurements.

In general, larger or more erratic COP shifts suggest instability or difficulty maintaining balance [27,28,29,35,36,37], as observed during static stance measurements of orthopedically diseased [28,29] and senior dogs (those over 75% into their expected lifespan) [33]. While conditions like cubarthrosis and gonarthrosis significantly increase MLD, CCD, and SS when compared to control groups [27,28], aging appears to have a minimal effect on CCD [33]. However, variations in measurement duration and repetitions (e.g., 3 × 20 s [27], 3 × 10 s [28], and 7 × 1 s [33]) complicate the interpretation of these findings [39]. While pressure and force plates yield valid and reliable measures of COP parameters, discrepancies in the literature can be found, particularly when different equipment [38] or measurement procedures [39] are used. According to the most recent literature, a single 10 s measurement or two repetitions of 5 s measurements provide optimal results for evaluating PS in dogs during static stance when using a pressure plate [39]. Once measurement procedures are standardized, more reliable results can be obtained so that when introducing further challenges to PS, more subtle and complex differences between adult and senior dogs might be identified.

The implementation of external mechanical perturbations on motorized platforms in the evaluation of PS is a novel approach in canine research. While researchers found a significant increase in COP parameters under perturbated conditions [30,40], their techniques require cost-intensive equipment and have not been used in diseased and senior animals yet. Static stance measurements with restricted sight are considerably easier to perform. This method has not been used in canine research up to date but has previously been performed in human [17,41,42] and equine [35] research. The effect of blindfolded posturography in healthy young adult humans has been studied with varying results [26,43]. While some researchers describe no effect on PS [26], others have found a significant decrease in COP displacement in the mediolateral and craniocaudal directions during blindfolded measurements compared to in the sighted condition, which was interpreted as an increase in muscular activity [43].

In contrast, a lack of visual information does not affect CCD in elderly humans while standing on a foam surface but results in a significant increase in MLD. Therefore, mediolateral PS is more dependent on visual input, while sagittal control is more affected by somatosensory input. When visual input is disturbed, the sensitivity to somatosensory information increases to maintain PS [17]. In contrast to young, healthy adults [26,43], in horses, the accuracy of PS is strongly dependent on visual input. Blindfolded posturographic measurements result in a significant increase in MLD, CCD, and mediolateral COP velocity compared to sighted measurements [35].

The impact of visual input on PS in dogs has not yet been investigated, and data on elderly dogs are scarce. However, based on the results of human research, differences between coping strategies in adult and senior dogs can be expected when faced with conditions challenging PS [17,43].

Thus, canine research should further focus on the effect of different measurement conditions on PS in both adult and senior dogs. Therefore, we conducted posturographic measurements during sighted and blindfolded quiet standing. Following previous research, the abbreviations for eyes open (EO) and eyes closed (EC) were selected [26]. We hypothesized that EC results in a significant increase in COP parameters in adult and senior dogs compared to EO, with more profound differences in senior dogs compared to adult dogs.

## 2. Materials and Methods

### 2.1. Approval and Consent

This study was approved by the Ethics and Animal Welfare Committee of the University of Veterinary Medicine, Vienna, in accordance with the University’s Good Scientific Practice guidelines and national legislation (ETK-148/10/2021).

### 2.2. Animals and Inclusion Criteria

This study included a total of 40 pet dogs, consisting of 20 adult dogs (G1 < 50% of fractional lifespan) and 20 senior dogs (G2 > 75% of fractional lifespan) [44,45]. The standing measurements of all 40 dogs were included in the data analysis, with the younger dogs selected to match the older dogs in weight and body height.

The inclusion criteria consisted of the absence of any clinical orthopedic, neurological, or visual diseases, and a minimum body mass of 10 kg to ensure a more uniform representation of PS, as smaller dogs may have different anatomical and biomechanical characteristics that could introduce variability in the results [33]. All dogs underwent a general clinical examination including visual gait assessment, orthopedic and neurologic examination, and objective gait analysis using a pressure measurement plate (FDM Type 2, Zebris Medical GmbH, Allgäu, Germany) [46]. Additionally, the owners were asked to fill out the Canine Cognitive Dysfunction Rating scale (CCDR) to assess the cognitive function of the animal. Dogs with a CCDR score > 49, which is the diagnostic threshold for the diagnosis of canine cognitive dysfunction syndrome [47], were not included.

The breeds of the dogs consisted of mixed breed, Labrador Retriever, Border Collie, Belgian Malinois, Flatcoated Retriever, Beagle, Standard Poodle, Irish Setter, Golden Retriever, Magyar Viszla, Greyster, Springer Spaniel, Rhodesian Ridgeback, and Coonhound. The mean age of the dogs in group 1 was 2.18 ± 1.29 years and 11.37 ± 1.23 years in group 2. The groups were matched by body mass, body height (measured from the dorsal border of the scapula), and body length (distance between Tuber supraglenoidale and Tuber ischiadicum) (Table 1).

### 2.3. Equipment and Measurement Procedure

The measurements were made for dogs during quiet standing on a pressure measurement plate (FDM Type 2, Zebris Medical GmbH, Allgäu, Germany) equipped with 15,360 sensors covering an area of 203 × 54.2 cm and a measuring frequency of 100 Hz. The sensor size of the platforms was 0.72 × 0.72 cm. The pressure plate was covered with a black 1 mm thick non-slip rubber mat. All measurement procedures were filmed using a Panasonic NV-MX500 camera (Panasonic, Kadoma, Osaka, Japan), as previously described [39].

### 2.4. Objective Gait Analysis

Prior to all procedures, the dogs were allowed to move freely in the room to get accustomed to the measurement setup. Afterward, the dogs were walked over the pressure plate until at least 5 valid passes for each paw were collected. A valid pass was defined as a walk in which the dogs crossed the plate in a straight line without changing their speed, turning their head, or pulling on the leash. A symmetric gait pattern was ensured by a symmetry index of peak vertical force (PFz) and vertical impulse (IFz) below 3%. The difference in the speed at which the dogs crossed the plate had to be within a range of ± 0.3 m/s and an acceleration of ± 0.5 m/s^2^ [39,46,48].

### 2.5. Static Posturography

After a short break, static posturography was performed during 2 conditions, including quiet standing with unrestricted sight and while wearing taped laser goggles (Laser Safety Doggles^®^, LASERVISION GmbH & Co.KG, 90,766 Fürth, Germany) (Figure 1). The dogs were asked to approach the plate using positive reinforcement. For the second condition, the laser goggles were put on after the dog found a comfortable standing position. The dogs had to stand still on the pressure measurement plate with all limbs perpendicular to the plate without any body, head, tail, limb, or paw movements.

For this purpose, the owner stood in front of the animal to maintain its attention during the measurement procedure. None of the dogs were wearing laser goggles before, which is why they were introduced using positive reinforcement methods. All dogs adapted to wearing them within a couple of minutes. Afterward, the dogs were first placed on the pressure plate, and then the goggles were carefully applied.

During the measurement, the owners were allowed to speak to their dogs, which further helped maintain their attention. The dogs did not show fear toward the goggles and were rewarded with treats when the goggles were both applied and removed to ensure positive reinforcement.

After each measurement, the animal was rewarded with a treat and asked to rest. As recommended, 2 valid passes of 5 s measurements per condition were collected and analyzed for each dog [39]. To ensure a static pose, the dogs were visually monitored in real time, and only sequences without movement were selected during post-experiment video analysis for the final evaluation.

### 2.6. Parameters Under Investigation

The fractional lifespan (FLS) was calculated using the following formula [44] adjusted from imperial to metric units [45]:(1)$$FLS=13.620+\left(0.0276\;body\times height\;in\;cm\right)-\left(0.1186\times body\;mass\;in\;kg\right)$$

All parameters were analyzed using a custom software Pressure Analyzer (Michael Schwanda, version 4.9.3.0), which was then exported to Microsoft Excel 2016. The following parameters were used for the evaluation of the inclusion criteria:The mean speed (m/s) and acceleration (m/s^2^) were calculated for the left forelimb;The symmetry index (SI), expressed as a percentage (SI%), was calculated for both parameters (PFz and IFz) according to the following equation modified from Budsberg et al. [49]:(2)$$SIXFz\left(\%\right)=abs\left(\frac{\left[XFzLLx-XFzRLx\right]}{\left[XFzLLx+XFzRLx\right]}\right)\times 100$$
where XFz is the mean value of the PFz or IFz of valid steps, LLx is the left fore- or hindlimb, and RLx is the right fore- or hindlimb. Perfect symmetry between the right and left fore- or hindlimbs was assigned a value of 0%.

For the posturographic analysis, the data were low-pass filtered using a fourth-order Butterworth Filter with a cutoff frequency of 10 Hz [50]. The following parameters were analyzed:

Base of support:Base of support (BOS)—area enclosed by the coordinates of the center of the paws, in cm^2^.Base of support length (BOS L)—distance between the center of the fore- and hindlimbs, in cm.Base of support width (BOS W)—distance between the center of the left and right limbs, in cm.

Center of pressure [27,28,29,30,31,32,33,34,35,36,37,38,39]:Craniocaudal displacement—Mean deviation on the craniocaudal axis. This was normalized to the BOS L and expressed as a percentage (CCD%).Mediolateral displacement—Mean deviation on the lateral axis. This was normalized to the BOS W and expressed as a percentage (MLD%).Statokinesiogram length—The length of the line that joins the points of the COP trajectory. This was normalized to the BOS and expressed as a percentage (L%).Support surface or statokinesiogram—The area determined by an ellipse that contains 90% of the points of the COP trajectory. This was normalized to the BOS and expressed as a percentage (SS%).Average speed (AS) (mm/s) of COP sway.

#### Romberg Index

The Romberg index (RI) is the ratio of the EC score to EO score multiplied by 100 (EC/EO × 100) [26,50]. It was calculated for each COP parameter and reflects the impact of the visual input and proprioceptive contribution to PS [26,51,52,53]. The RI was calculated for each COP parameter as follows:(3)$$RI=\frac{standing\;measurement\;with\;EC}{standing\;measurement\;with\;EO}\times 100$$

### 2.7. Statistical Analysis

All statistical analyses of the standing measurements of 40 dogs were performed using IBM SPSS v27. The effects of different measurement conditions and groups on the parameters were analyzed using linear mixed-effects models in which the conditions and groups were added as fixed factors to the model. Sidak’s alpha correction was applied for multiple comparisons. The assumption of a normal distribution was tested using the Shapiro–Wilk test. For all analyses, a *p*-value below 5% (*p* < 0.05) was seen as significant.

## 3. Results

### 3.1. Base of Support

Descriptive statistics are displayed in Appendix A for each group and measurement condition. No significant difference was found in the BOS parameters between the groups (Appendix A) and measurement conditions (Appendix A).

### 3.2. Center of Pressure

Descriptive statistics of the conditions EO and EC for all COP parameters in adult and senior dogs can be found in Table 2 and are illustrated in Figure 2.

The loss of visual input led to a significant decrease in CCD% and AS in adult dogs, but had no impact on MLD%, L%, and SS%. No significant difference was found in any COP parameters between the conditions in senior dogs (Table 3, Figure 2).

No significant difference in COP parameters between groups was found with EO. During the blinded conditions, CCD% and SS% in senior dogs showed significantly higher values compared to in adult dogs. The remaining COP parameters did not show a significant difference between groups (Table 4, Figure 2).

### 3.3. Romberg Index

The mean and standard deviation and *p*-values of the group comparison of the RIs for each COP parameter are summarized in Table 5. In both groups, the RIs of MLD% and SS% exceeded 100, while CCD% exceeded 100 only in senior dogs. Additionally, the RI of CCD% was significantly higher in senior dogs compared to in adult dogs.

## 4. Discussion

This study aimed to evaluate the influence of visual input and aging on PS in dogs by analyzing conventional COP parameters during static posturographic measurement under sighted and blindfolded conditions. The findings reveal critical insights into the sensory and motor mechanisms underlying balance in adult and senior dogs and their adaptations to visual deprivation. The hypotheses were that the condition EC results in a significant increase in COP parameters in adult and senior dogs compared to EO, with more profound differences in senior dogs compared to adult dogs. This was partially confirmed.

Surprisingly, a significant decrease in CCD% in adult dogs with EC was observed when compared to EO, which was not evident in senior dogs. Likewise, AS was significantly reduced during the blindfolded condition in adult dogs, with, again, no significant effect in senior dogs. These changes suggest that visual deprivation prompts adult dogs to rely more heavily on somatosensory input, particularly for sagittal stability. While a reduction in COP parameters during challenging conditions is hardly mentioned in canine postural research, healthy young adults show a similar pattern during blindfolded conditions, which was interpreted as an increase in muscular activity [43].

Conversely, senior dogs showed no significant changes in COP parameters when blindfolded, highlighting their baseline dependency on visual input for stability. While adult dogs cope with the loss of visual input by reducing the extent of postural sway, we propose that senior dogs are unable to use this mechanism anymore. This aligns with findings in elderly humans, where visual input is critical for stability. The inability of senior dogs to compensate for the loss of visual input by reducing COP sway, as seen in young adult dogs, may reflect an age-related decline in sensory integration or diminished adaptability in motor responses. When visual input is disturbed, the sensitivity to somatosensory information increases to maintain PS [17].

In contrast to previous research [33], a static stance without further challenges to PS did not result in significant differences between healthy adult and healthy senior dogs in any of the measured COP parameters. We assume that these discrepancies might be the result of differences in measurement durations (7 × 1 s [30] compared to 2 × 5 s). Indeed, based on a recent validation study, 1 s measurements produced the lowest reliability of COP parameters [39].

Age-related differences became evident under blindfolded conditions, with senior dogs exhibiting significantly higher values for CCD% and SS% compared to adult dogs, indicating decreased balance control and a less stable stance. Such differences were previously reported during stance measurements in dogs with cubarthrosis and gonarthrosis, showing a significant increase in MLD, CCD, and SS when compared to control groups [27,28].

Interestingly, MLD% did not differ significantly between groups, even under blindfolded conditions. This suggests either that mediolateral stability may not deteriorate as profoundly as sagittal stability in aging dogs or that the deprivation of visual input is not a proper method to provoke instability in the transversal axis. The latter theory contrasts with human research. Mediolateral PS in elderly humans is more dependent on visual input, while sagittal control is more affected by somatosensory input, like standing on a foam surface [17]. However, due to anthropometric differences between humans as bipeds and quadrupedal species, a direct comparison is not possible.

In addition to MLD%, L% did not differ significantly between groups or conditions. One possible explanation for this could be that the statokinesiogram length is influenced by the general ability to maintain stability rather than specific postural control mechanisms [54]. Since both groups were able to maintain a relatively consistent trajectory despite age-related differences, it is plausible that factors such as the overall stability of the dogs, irrespective of age, resulted in no significant variation in L%. This stability may not have been sufficiently challenged by the conditions, as was the case with MLD%, where visual input deprivation did not have a pronounced effect.

The RI provides a quantitative measure of the contribution of visual input to PS. It represents the percentage difference between the results of static measurement during the EC condition compared to the EO condition. A value above 100 indicates an increase in values during the blindfolded measurement condition, which is interpreted as a high reliance on visual information and a low proprioceptive contribution to the parameter [23,49,51]. In this study, senior dogs exhibited a significantly higher RI for CCD% compared to adult dogs, indicating a greater reliance on visual input for sagittal stability. Interestingly, the RIs for CCD%, MLD%, and SS% exceeded 100 in both groups, underscoring a dominant role of visual input in mediolateral stability. These findings are consistent with equine studies, where craniocaudal and mediolateral control is sensitive to visual deprivation [35]. However, visual deprivation alone did not challenge PS in adult or senior dogs to the point where MLD% showed a significant difference between conditions or groups.

These findings emphasize the differential roles of visual and somatosensory systems in maintaining PS across age groups. In adult dogs, sensory integration appears more dynamic, allowing for compensatory adjustments in the absence of visual cues. In contrast, senior dogs exhibit a more rigid dependence on visual input, potentially due to age-related declines in proprioceptive function or slower central processing of sensory information. The results underscore the importance of visual input in maintaining PS, particularly for senior dogs, and highlight the reduced capacity for sensory reweighting with age. These findings align with the broader literature on aging and postural control, emphasizing the need for targeted interventions, such as proprioceptive training, to improve sensory integration and mitigate injuries and fall risk in older populations [55].

Compensatory stepping to increase the BOS is a well-known coping mechanism in challenging conditions to prevent falling, which has been described in humans and dogs [14,30]. However, aging affects proprioceptive and tactile sensitivity, particularly in the plantar region, resulting in the impaired control of compensatory stepping [18]. Since no significant differences were found in BOS parameters between conditions or groups, this coping mechanism either is not triggered by the loss of visual input alone or is hindered by the tactile loss due to aging.

Aging dogs face an increased risk of falls due to limb weakness and neuromuscular degeneration, though their quadrupedal stance and proximity to the ground reduce the severity of fall-related injuries compared to humans. The mechanisms of aging and functional decline in dogs closely resemble those in humans [17,18,56], making them a valuable model for studying frailty and intervention strategies. As research progresses, validating clinical assessment tools will be essential for the early detection of mobility issues and the development of therapeutic strategies aimed at extending the healthy lifespan and quality of life of aging dogs. Addressing risk factors such as obesity and inactivity, alongside adapting exercise-based interventions from human medicine, will be critical in mitigating functional decline and preserving quality of life in aging dogs [56].

While this study provides valuable insights into canine PS, several limitations should be noted. The relatively small sample size may limit the generalizability of the findings. Additionally, breed differences in COP parameters, although minimized through careful group matching, could introduce variability. Since different breeds have varying average lifespans, using multiple breeds may affect result accuracy. To address this, the fractional lifespan was calculated based on the height and body mass of the dogs to provide a more standardized measure of aging [44,45]. Future studies should consider focusing on a single breed to enhance result credibility or employing a larger sample size to account for breed-related differences.

Furthermore, the position of the owner during the static measurement could influence the results. The owner stood in front of the dog to maintain its attention during the procedure, which may have affected the results. Future research could explore the impact of the owner’s position on standing measurements to determine whether this factor plays a significant role.

## 5. Conclusions

This study demonstrates the crucial role of visual input in PS and reveals significant age-related differences in sensory integration among dogs. While no major disparities were observed between adult and senior dogs under normal sighted conditions, blindfolding exposed key differences. Senior dogs exhibited a greater CCD% and increased SS%, suggesting a stronger reliance on vision for sagittal stability. In contrast, adult dogs displayed reduced postural sway under blindfolded conditions, indicating a compensatory shift toward somatosensory reliance. These findings highlight diminished sensory adaptability with aging, which may contribute to balance impairments in senior dogs. Recognizing these age-specific sensory adaptations is crucial for developing targeted interventions, such as balance training, to enhance sensory integration and mobility in aging dogs. Future research should explore dynamic and multimodal postural challenges to further elucidate compensatory mechanisms and refine clinical applications for improving canine quality of life.

## Figures and Tables

**Figure 1 sensors-25-01300-f001:**
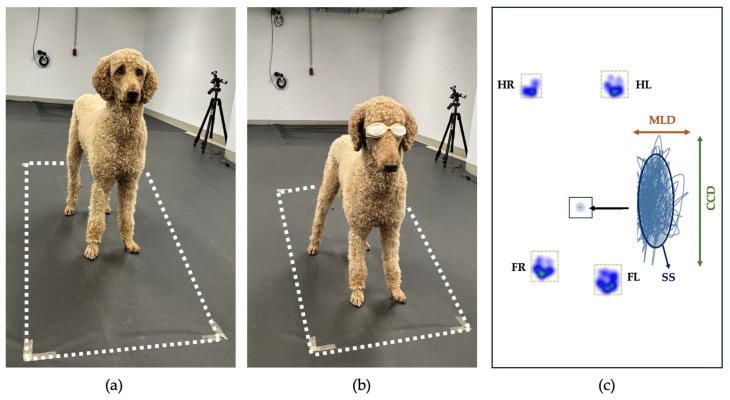
Experimental setup: The pressure measurement plate (dotted white line) is hidden under a non-slip rubber mat to ensure an undisturbed standing position. The dogs stand with all paws on the plate with unrestricted sight (EO) (**a**) and while wearing taped laser goggles (Laser Safety Doggles^®^, LASERVISION GmbH & Co.KG, 90,766 Fürth, Germany) (EC) (**b**). The body COP (black square) is measured between the limbs: front right (FR), front left (FL), hind right (HR), and hind left (HL). The blue trajectory represents a magnified view of the COP trajectory. Within this, the craniocaudal displacement (CCD, green arrow), mediolateral displacement (MLD, orange arrow), and support surface (SS, blue ellipse) are indicated (**c**).

**Figure 2 sensors-25-01300-f002:**
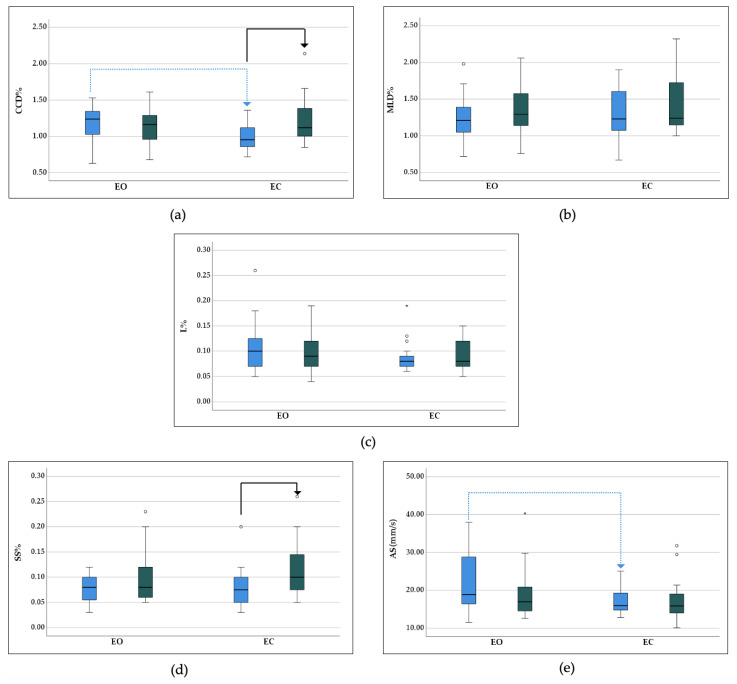
Pairwise comparison of the effect of measurement conditions, including standing with eyes open (EO) and with taped laser goggles (EC), in adult (blue) and senior (green) dogs on COP parameters: (**a**) craniocaudal displacement (CCD%), (**b**) mediolateral displacement (MLD%), (**c**) statokinesiogram length (L%), (**d**) support surface (SS%), and (**e**) average speed (AS) of adult (G1) and senior (G2) dogs during sighted (EO) and blindfolded (EC) measurements. Significant differences between groups are marked with a black arrow, and significant differences between conditions are marked with a blue dotted arrow (*p* < 0.05). The stars and circles represent statistical outliers.

**Table 1 sensors-25-01300-t001:** Demographics of the participating dogs in group 1 (adult) and group 2 (senior), including mean ± SD of body mass, height, length, and CCDR score.

Group	Body Mass (kg)	Height (cm)	Length (cm)	CCDR
G1	21.87 ± 5.66	54.27 ± 6.78	57.08 ± 5.78	34.50 ± 1.24
G2	22.34 ± 6.37	52.90 ± 6.63	55.81 ± 6.58	36.30 ± 4.43

**Table 2 sensors-25-01300-t002:** Descriptive statistics (mean ± SD) of the COP parameters, including craniocaudal displacement (CCD%), mediolateral displacement (MLD%), statokinesiogram length (L%), support surface (SS%), and average speed (AS), of adult (G1) and senior (G2) dogs during sighted (EO) and blindfolded (EC) measurements.

Condition	Group	CCD%	MLD%	L%	SS%	AS (mm/s)
EO	G1	1.18 ± 0.18 ^#^	1.23 ± 0.35	0.11 ± 0.03	0.08 ± 0.04	21.90 ± 3.33 ^#^
	G2	1.14 ± 0.30	1.37 ± 0.43	0.10 ± 0.03	0.10 ± 0.05	19.18 ± 5.46
EC	G1	0.99 ± 0.23 *^#^	1.30 ± 0.32	0.09 ± 0.05	0.08 ± 0.03 *	17.08 ± 7.69 ^#^
	G2	1.22 ± 0.27 *	1.46 ± 0.35	0.09 ± 0.03	0.11 ± 0.05 *	17.18 ± 7.00

* Significant difference between groups in each column; ^#^ significant difference within group between conditions in each column (*p* < 0.05).

**Table 3 sensors-25-01300-t003:** *p*-values of the comparison of the effect of blindfolding in group 1 (adult dogs) and group 2 (senior dogs) for the COP parameters, including craniocaudal displacement (CCD%), mediolateral displacement (MLD%), statokinesiogram length (L%), support surface (SS%), and average speed (AS), of adult (G1) and senior (G2) dogs.

Group	CCD%	MLD%	L%	SS%	AS
G1	0.019	0.531	0.074	0.917	0.015
G2	0.307	0.466	0.466	0.349	0.305

**Table 4 sensors-25-01300-t004:** *p*-values of the comparison of group 1 (adult dogs) and group 2 (senior dogs) during sighted (EO) and blindfolded (EC) measurements for the COP parameters, including craniocaudal displacement (CCD%), mediolateral displacement (MLD%), statokinesiogram length (L%), support surface (SS%), and average speed (AS), of adult (G1) and senior (G2) dogs.

Condition	CCD%	MLD%	L%	SS%	AS
EO	0.647	0.192	0.438	0.180	0.248
EC	0.005	0.223	0.833	0.032	0.947

**Table 5 sensors-25-01300-t005:** Descriptive statistics (mean ± SD) and *p*-values of RI of craniocaudal displacement (CCD%), mediolateral displacement (MLD%), statokinesiogram length (L%), support surface (SS%), and average speed (AS) of adult (G1) and senior (G2) dogs.

Group	RI CCD%	RI MLD%	RI L%	RI SS%	RI AS
G1	87.31 ± 23.39 *	111.33 ± 36.83	87.84 ± 26.19	107.58 ± 55.99	84.67 ± 24.71
G2	109.75 ± 23.58 *	109.22 ± 29.58	93.60 ± 25.74	121.87 ± 37.05	93.21 ± 22.83
*p*-value	0.004	0.843	0.487	0.347	0.263

* Significant differences between groups (*p* < 0.05).

## Data Availability

The original contributions presented in this study are included in the article. Further inquiries can be directed to the corresponding author.

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
