# Peer review of "Impact of Aging and Visual Input on Postural Stability in Dogs: Insights from Center-of-Pressure Analysis"

_sensors, 2025, doi:10.3390/s25051300_

Round 1

Reviewer 1 Report

Comments and Suggestions for Authors

The review report

The manuscript title: Impact of Aging and Visual Input on Postural Stability in Dogs: Insights from Center of Pressure Analysis.

In this manuscript, the authors investigate the effect of visual input and aging on postural stability in dogs by analyzing center of pressure parameters during static center of pressure imaging under blindfold and center of pressure conditions. Twenty adult dogs without any orthopedic, neurological, or visual impairment were included in the study using a pressure plate. The results showed that senior dogs rely more on visual input for sagittal stability. On the other hand, adult dogs show an adaptive shift toward a greater reliance on somatosensory input. The study demonstrated poor sensory integration and adaptive capacity in senior dogs, which the authors interpret as an age-related decline in proprioception and sensory processing. Finally, the authors recommend that targeted interventions such as balance training should be adopted to enhance sensory integration and reduce the risk of falls in senior dogs.

The authors have used references relevant to the topic of their article, and the conclusions presented are consistent with the evidence and arguments presented, and address the main question raised. However, this article can be accepted for publication after taking into account the following points:

1. All keywords must start with a capital letter.

2. Authors should explain how the fractional lifespan was calculated using appropriate references.

3. Line 114. Why were the same number of dogs and age requirements used in Reference 44 adopted?

4. Line 135. It is preferable to write, “The measurements were made for dogs” instead of “All dogs were measured”.

5. Lines 174 and 204. Each equation must be numbered and its source must be cited.

6. The comparison method used in Figure 2 is not appropriate. Discussing each case separately and drawing a figure for each case will facilitate and enrich the comparison process.

7. The Abstract is richer in results than the conclusion, contrary to what it should be. The conclusion is too short and poor in results, and in its current form affects the uniqueness of the results, as it presents obvious results. The conclusion includes more recommendations than results, and it should be reformulated.

8. I believe that using more than one breed of dog will affect the accuracy of the results. Conducting the experiments on one breed will give more credibility to the results especially since each breed has a different average lifespan.

Author Response

Dear Reviewer 1,

Thank you for your thorough review and insightful comments on our manuscript. We appreciate your positive feedback on the relevance of our references and the consistency between our conclusions and the evidence presented. Your input will certainly enhance the clarity and depth of our manuscript. Please find the detailed responses below and the corresponding revisions highlighted in yellow in the re-submitted files.

  1. All keywords must start with a capital letter.

- We corrected all keywords.

  1. Authors should explain how the fractional lifespan was calculated using appropriate references.

- Done, we added the reference, and the equation used to calculate the fractional lifespan.

Line 125-126: “This study included a total of 40 pet dogs, consisting of 20 adult dogs (G1 < 50% of fractional lifespan) and 20 senior dogs (G2 > 75% of fractional lifespan) [44,45].”

Line 198-199: “The fractional lifespan (FLS) was calculated using the following formula [44] adjusted from imperial to metric units [45]:

FLS=13.620+(0.0276 body×height in cm)-(0.1186×body mass in kg)      (1)”

  1. Line 114. Why were the same number of dogs and age requirements used in Reference 44 adopted?

- Thank you for your question. We matched the younger dogs to the older dogs in terms of weight and body height to enhance comparability and control for potential size-related effects.

Line 126-128: “The standing measurement of all 40 dogs were included in the data analysis, with the younger dogs selected to match the older dogs in weight and body height.”

  1. Line 135. It is preferable to write, “The measurements were made for dogs” instead of “All dogs were measured”.

- Done, line 150: “The measurements were made for dogs during quiet standing on a pressure measure-ment plate (FDM Type 2, Zebris Medical GmbH, Allgäu, Germany) equipped with 15,360 sensors covering an area of 203 × 54.2 cm and a measuring frequency of 100 Hz.”

  1. Lines 174 and 204. Each equation must be numbered and its source must be cited.

- We added the numeration and references for the equations and parameters.

  1. The comparison method used in Figure 2 is not appropriate. Discussing each case separately and drawing a figure for each case will facilitate and enrich the comparison process.

- Thank you for your feedback. We believe that the chosen comparison method in Figure 2 is appropriate for our study, as it allows for a clear and concise presentation of the data. Additionally, the approach was well received by the other reviewers. Regarding the suggested alternative presentation, we respectfully decline this change, as none of the other four reviewers requested it, and the number of figures required would not justify restructuring the data in this way. We believe our current approach provides a balanced and effective visualization of the results. Furthermore, we have now labeled the individual panels in Figure 2 with (a) to (e) and hope that this adjustment will make the figure easier to follow and understand.

  1. The Abstract is richer in results than the conclusion, contrary to what it should be. The conclusion is too short and poor in results, and in its current form affects the uniqueness of the results, as it presents obvious results. The conclusion includes more recommendations than results, and it should be reformulated.

- Thank you for your valuable feedback. We have revised the conclusion to better reflect the richness of our results, ensuring it emphasizes key findings rather than focusing too much on recommendations. The updated version highlights the observed age-related differences in postural stability and sensory reliance, aligning more effectively with the study's contributions.

Line 403-416: “This study demonstrates the crucial role of visual input in PS and reveals significant age-related differences in sensory integration among dogs. While no major disparities were observed between adult and senior dogs under normal sighted conditions, blind-folding exposed key differences. Senior dogs exhibited greater CCD% and increased SS%, suggesting a stronger reliance on vision for sagittal stability. In contrast, adult dogs displayed reduced postural sway under blindfolded conditions, indicating a compen-satory shift towards somatosensory reliance. These findings highlight diminished sensory adaptability with aging, which may contribute to balance impairments in senior dogs. Recognizing these age-specific sensory adaptations is crucial for developing targeted interventions, such as balance training, to enhance sensory integration and mobility in aging dogs. Future research should explore dynamic and multimodal postural challenges to further elucidate compensatory mechanisms and refine clinical applications for im-proving canine quality of life.”

  1. I believe that using more than one breed of dog will affect the accuracy of the results. Conducting the experiments on one breed will give more credibility to the results especially since each breed has a different average lifespan.

- Thank you for the insightful feedback. Breed differences, particularly in lifespan, may influence postural stability results. To improve standardization, the fractional lifespan was calculated based on height and body weight. The limitations section has been revised to reflect this consideration and suggest future studies focus on a single breed or a larger sample size.

Line 391-397: “Additionally, breed differences in COP parameters, although minimized through careful group matching, could introduce variability. Since different breeds have varying average lifespans, using multiple breeds may affect result accuracy. To address this, the fractional lifespan was calculated based on the height and body mass of the dogs to provide a more standardized measure of aging [44,45]. Future studies should consider focusing on a single breed to enhance result credibility or employing a larger sample size to account for breed-related differences.”

Reviewer 2 Report

Comments and Suggestions for Authors

The aim of this study was to determine how balance in dogs is affected by age, using force plate analysis. Although no differences were found based on age when dogs were not blindfolded, postural stability was reduced in older dogs when blindfolded. The authors conclude that older dogs rely less on somatosensory input and more on visual input for balance, indicating a need for balance training to improve sensory integration and reduce fall risk.

This is a well-controlled study with a decent sample size (two groups of N=20 dogs). The article is well-structured and follows a logical flow. The experimental design is sound for testing the hypotheses. One key issue with the current state of the paper is that it does not sufficiently describe the clinical significance of postural stability in dogs.

Specific Comments:

Pg 2 line 84: I would reword “has been discussed controversially” to “has been studied with varying results” (or similar), and also clarify that “healthy young adults” is referring to humans.

The background section should discuss the importance of balance (and measuring balance) in dogs, including the prevalence and consequences of falls, with appropriate references.

pg. 3 line 116: Explain reasoning for minimum body mass criteria.

Pg. 4 line 157: What method(s) were used to ensure no movement of body, head, tail, limb, and paws? Purely visual in real-time, or did you incorporate video analysis to confirm static poses after the experiments?

Pg. 4 lines1 157-159: How was the dog’s attention maintained during the blindfolded state? Also, did the goggles induce any fear and erratic movement in the dogs, and how was this mitigated?

Pg. 9 lines 301-307: Here you discuss possible explanations for MLD% not being affected, but what about L%, which also didn’t differ between groups or conditions? Add short discussion about why L% may have remained constant.

Figure 2: It’s a little confusing how you added a note using an asterisk, but it doesn’t correspond to the asterisks on the plot; please use a different symbol for the note and also explain asterisks and circles within the plot.

Pg. 8 lines 255-256: Sentence starting with “RI” is confusing and may need to be reworded.

Author Response

Dear Reviewer 2,

Thank you for your thoughtful and constructive feedback on our manuscript. We are grateful for your positive comments regarding the study design, structure, and logical flow of the article. We appreciate your recognition of the soundness of the experimental approach. We also acknowledge your suggestion about the need for a more detailed discussion of the clinical significance of postural stability in dogs. Please find the detailed responses below and the corresponding revisions highlighted in green in the re-submitted files.

Specific Comments:

Pg 2 line 84: I would reword “has been discussed controversially” to “has been studied with varying results” (or similar), and also clarify that “healthy young adults” is referring to humans.

- Thank you for your suggestion, we adjusted the sentence (Line 94-96):
“The effect of blindfolded posturography in healthy young adult humans has been studied with varying results.”

The background section should discuss the importance of balance (and measuring balance) in dogs, including the prevalence and consequences of falls, with appropriate references.

- Thank you for your comment. We have revised the background section to emphasize the importance of balance and postural stability in dogs, especially in aging animals. We included a paragraph on how weakened muscles, balance issues, and age-related conditions can increase the risk of falls in older dogs, along with the potential health consequences, even though falls in dogs may not lead to the same severe injuries seen in humans.

Background: Line 52-61: “In humans, falls are a major health concern, often leading to severe injuries due to de-clining physical function. As a result, fall prevention is a key area of focus in both clinical research and intervention strategies [17,18]. Similarly, aging dogs face an increased risk of falls due to muscle weakness, balance deficits, and age-related conditions such as musculoskeletal and neurological disorders. However, because dogs are quadrupeds and have a lower center of gravity, they are less likely to sustain serious injuries from falls compared to humans. Despite this reduced injury risk, maintaining PS in senior dogs is still crucial, as falls can negatively impact their overall health and mobility. Therefore, assessing and addressing balance impairments in aging dogs is an important aspect of veterinary care to help preserve their quality of life [19].”

Discussion: Line 379-388: “Aging dogs face an increased risk of falls due to limb weakness and neuromuscular degeneration, though their quadrupedal stance and proximity to the ground reduce the severity of fall-related injuries compared to humans. The mechanisms of aging and functional decline in dogs closely resemble those in humans [17,18,56], making them a valuable model for studying frailty and intervention strategies. As research progresses, validating clinical assessment tools will be essential for early detection of mobility issues and the development of therapeutic strategies aimed at extending the healthy lifespan and quality of life of aging dogs. Addressing risk factors such as obesity and inactivity, alongside adapting exercise-based interventions from human medicine, will be critical in mitigating functional decline and preserving quality of life in aging dogs [56].”

  1. 3 line 116: Explain reasoning for minimum body mass criteria.

-  Thank you for your suggestion; we limited our study to dogs above 10 kg to reduce variability related to the anatomical and biomechanical differences seen in smaller dogs. This information was added to materials and methods.

Line 129-132: “The inclusion criteria consisted of the absence of any clinical orthopedic, neurological, or visual diseases, and a minimum body mass of 10 kg to ensure a more uniform representation of postural stability, as smaller dogs may have different anatomical and biomechanical characteristics that could introduce variability in the results [33].”

Pg. 4 line 157: What method(s) were used to ensure no movement of body, head, tail, limb, and paws? Purely visual in real-time, or did you incorporate video analysis to confirm static poses after the experiments?

- Thank you for your comment. To ensure no movement of the body, head, tail, limbs, or paws, we visually monitored the dogs in real-time while they were standing on the pressure plate, and the entire measurement process was filmed. During post-experiment analysis, we carefully selected only the sequences without movement for evaluation of static poses. This information was added to materials and methods.

Line 185-187: “To ensure a static pose, the dogs were visually monitored in real-time, and only sequences without movement were selected during post-experiment video analysis for the final evaluation.”

Pg. 4 lines1 157-159: How was the dog’s attention maintained during the blindfolded state? Also, did the goggles induce any fear and erratic movement in the dogs, and how was this mitigated?

- Thank you for your comment. The dogs were first placed on the pressure plate, and then the goggles were carefully applied. During the measurement, the owners were allowed to speak with their dogs, which helped maintain their attention. The dogs did not show fear towards the goggles and were rewarded with treats both when the goggles were applied and removed to ensure positive reinforcement

Line 174-182: “For this purpose, the owner stood in front of the animal to maintain its attention during the measurement procedure. None of the dogs were wearing laser googles before, which is why they were introduced using positive reinforcement methods. All dogs adapted to wearing them within a couple of minutes. Afterwards, the dogs were first placed on the pressure plate, and then the goggles were carefully applied.

During the measurement, the owners were allowed to speak with their dogs, which further helped maintain their attention. The dogs did not show fear towards the goggles and were rewarded with treats both when the goggles were applied and removed to ensure positive reinforcement.”

Pg. 9 lines 301-307: Here you discuss possible explanations for MLD% not being affected, but what about L%, which also didn’t differ between groups or conditions? Add short discussion about why L% may have remained constant.

- Thank you for your comment. In addition to the explanations for MLD%, we also considered that L% might remain constant due to its dependence on overall stability rather than specific postural control mechanisms. Both groups were able to maintain a consistent COP trajectory, which may explain the lack of significant differences in L%, even under the condition tested.

Line 341-348: “In addition to MLD%, L% did not differ significantly between groups or conditions. One possible explanation for this could be that the statokinesiogram length is influenced by the general ability to maintain stability rather than specific postural control mechanisms [54]. Since both groups were able to maintain a relatively consistent trajectory despite age-related differences, it is plausible that factors such as the overall stability of the dogs, irrespective of age, resulted in no significant variation in L%. This stability may not have been sufficiently challenged by the conditions, as was the case with MLD%, where visual input deprivation did not have a pronounced effect.”

Figure 2: It’s a little confusing how you added a note using an asterisk, but it doesn’t correspond to the asterisks on the plot; please use a different symbol for the note and also explain asterisks and circles within the plot.

- Thank you for your comment. The asterisks in the figure were not used to mark any specific data points but represent outliers.

Pg. 8 lines 255-256: Sentence starting with “RI” is confusing and may need to be reworded.

- The sentence was reworded.

Line 285-287: “In both groups, the RI of MLD% and SS% exceeded 100, while CCD% exceeded 100 only in senior dogs. Additionally, the RI of CCD% was significantly higher in senior dogs, compared to adult dogs.”

Reviewer 3 Report

Comments and Suggestions for Authors

The article deals with an brand new approach to the application of force platforms to monitor the stability of animal posture in dogs throughout the animal's age. 

Nevertheless although the methodology is very well founded and the statistical analyzes of the results are also clearly processed, providing support for the conclusions, I felt that there was a lack of greater detail on the raw results of the use of the force platform, before the statistical analyses.

My suggestion is to present the intermediate result containing the raw maps (see attached pdf file as an example). This could have a lot to contribute to the article, specifically to help researchers who want to reproduce the proposed method for this type of application.

Author Response

Dear Reviewer 3,

Thank you for your valuable feedback. We appreciate your suggestion and understand the importance of presenting raw results before statistical analyses. To address this, we have added a corresponding figure (Figure 1, page 5) to illustrate key aspects of our methodology. We hope this addition enhances clarity and supports researchers in reproducing our approach.

Reviewer 4 Report

Comments and Suggestions for Authors

Slightly more detailed explanation (or different visualization) of the COP parameters would assist in the understanding of the results for readers without an in-depth understanding of these measurements.  However, with the available referencing the current description is sufficient as they allow readers the opportunity to further explore the topic if required.    

Author Response

Dear Reviewer 4,

Thank you for your thoughtful suggestion. We truly appreciate your recommendation to provide a more detailed explanation or alternative visualization of the COP parameters. We added a figure explaining presenting the COP parameters than can be visualized (Figure 1, page 5). Further, we hope that the revisions made in response to the other reviewers' comments will help enhance the clarity and understanding of the content for all readers. We trust these changes will offer a clearer foundation for those interested in further exploring the topic.

Reviewer 5 Report

Comments and Suggestions for Authors

The research focuses on the impact of aging and visual input on Postural Stability (PS) in Dogs by evaluating the center of pressure (COP). The following comments should be considered to improve the paper's quality.

1. The inclusion criterion:

- In lines 114–123, both inclusion and exclusion criteria are described. The authors should clarify whether all 40 dogs (20 adults and 20 seniors) mentioned were included in the study or if the 40 dogs represent a subset of a larger pool from which participants were selected.

- The authors could clarify whether all dogs have been asked to wear any laser goggles before or not. This might influence their behavior during measurements.

2. (Line 158) When EC, "...the owner stood in front of the animal to maintain its attention during the measurement procedure."  Please add a discussion about the position of the owner and whether it affects the mediolateral displacement or not. 

3. The Supplementary consists of only three Tables; the authors should include these Tables in the manuscript for ease to follow.

4. The equations should be numbered (Lines 174, 204).

5. Typos and presentation

- A brief legend explaining why RI values above 100 are significant at line number 309 should be mentioned after the equation of RI at line 204 for easy to follow.

-  Using of unit: Consider the unique unit for the length (convert cm => m)

- The Figures must be provided with high-quality images.

- Figure 2: Caption must move below the Figure and add (a), (b), etc to each subfigure.

- Line 155: laser google => goggle

- Line 174, 204: should use multiplication symbol instead of star or "x" in the equation.

After minor revisions, I recommend the paper be accepted for publication in Sensors.

Author Response

Dear Reviewer 5,

Thank you for your thorough review and valuable suggestions regarding our manuscript. We appreciate your constructive comments, especially your insights on the inclusion criteria, the impact of the owner's position during measurements, and the clarity of our results. Your feedback has greatly contributed to improving the precision and readability of our work. Please find the detailed responses below and the corresponding revisions highlighted in pink in the re-submitted files.

  1. The inclusion criterion:

- In lines 114–123, both inclusion and exclusion criteria are described. The authors should clarify whether all 40 dogs (20 adults and 20 seniors) mentioned were included in the study or if the 40 dogs represent a subset of a larger pool from which participants were selected.

- Thank you for your comment. Indeed, all 40 dogs were included in the study. To make this clear, we made the following changes:

Line 125-127: “This study included a total of 40 pet dogs, consisting of 20 adult dogs (G1 < 50% of fractional lifespan) and 20 senior dogs (G2 > 75% of fractional lifespan) [44,45]. The standing measurement of all 40 dogs were included in the data analysis, with the younger dogs selected to match the older dogs in weight and body height.”

Line 239-240: “All statistical analyses of the standing measurements of 40 dogs were performed using IBM SPSS v27.”

- The authors could clarify whether all dogs have been asked to wear any laser goggles before or not. This might influence their behavior during measurements.

- Thank you for your comment. Prior to the blindfolded measurement, all dogs were introduced to the laser goggles in a positive manner, as they had no prior experience with them. We added this description to the manuscript:
Line 175-178: “None of the dogs were wearing laser googles before, which is why they were introduced using positive reinforcement methods. All dogs adapted to wearing them within a couple of minutes. Afterwards, the dogs were first placed on the pressure plate, and then the goggles were carefully applied.”

  1. (Line 158) When EC, "...the owner stood in front of the animal to maintain its attention during the measurement procedure."  Please add a discussion about the position of the owner and whether it affects the mediolateral displacement or not. 

- Thank you for this suggestion. Indeed, we are not aware of research regarding the influence of the standing position of the owner during static measurements. Of course, this could have an effect on COP parameters. Therefore, we added the following limitation:

Line 398-402: “Further, the position of the owner during the static measurement could influence the results. The owner stood in front of the dog to maintain its attention during the procedure, which may have affected the results. Future research could explore the impact of the owner's position on standing measurements to determine whether this factor plays a significant role.”

  1. The Supplementary consists of only three Tables; the authors should include these Tables in the manuscript for ease to follow.

- Thank you for this suggestion. We are happy to include the supplements in the main manuscript, but none of the other 4 reviewers asked for them to be in the main text. This is why we would like to leave the final decision with the editor.

  1. The equations should be numbered (Lines 174, 204).

- Done.

  1. Typos and presentation

- A brief legend explaining why RI values above 100 are significant at line number 309 should be mentioned after the equation of RI at line 204 for easy to follow.

- Thank you for your comment! We have added the following description and hope that it enhances the understandability.
Line 349-354: “The RI provides a quantitative measure of the contribution of visual input to PS. It represents the percentage difference between the results of the static measurement during the EC condition compared to the EO condition. A value above 100 indicates an increase in the values during the blindfolded measurement condition, which is interpreted as a high reliance on visual information and a low proprioceptive contribution to the pa-rameter [23, 49, 51].”

-  Using of unit: Consider the unique unit for the length (convert cm => m)

- Thank you for your comment. Since publications concerning COP measurements usually use centimeters as the unit, we prefer to keep it this way to make it more easily comparable.

- The Figures must be provided with high-quality images.

- We updated both figures and send it as high-quality images.

- Figure 2: Caption must move below the Figure and add (a), (b), etc to each subfigure.

- Thank you for this suggestion, we updated Figure 2.

- Line 155: laser google => goggle

- corrected

- Line 174, 204: should use multiplication symbol instead of star or "x" in the equation.

- corrected

Round 2

Reviewer 1 Report

Comments and Suggestions for Authors

The authors have completed the required revisions, so I suggest accepting the article for publication.